# The Role of Breast Cancer Stem Cell-Related Biomarkers as Prognostic Factors

**DOI:** 10.3390/diagnostics10090721

**Published:** 2020-09-19

**Authors:** Clarence Ching Huat Ko, Wai Kit Chia, Gayathri Thevi Selvarajah, Yoke Kqueen Cheah, Yin Ping Wong, Geok Chin Tan

**Affiliations:** 1Department of Pathology, Faculty of Medicine, Universiti Kebangsaan Malaysia, Jalan Yaacob Latif, Bandar Tun Razak, 56000 Kuala Lumpur, Malaysia; clarence@ppukm.ukm.edu.my (C.C.H.K.); wk_chia@ppukm.ukm.edu.my (W.K.C.); 2Department of Biomedical Science, Faculty of Medicine & Health Sciences, Universiti Putra Malaysia, 43400 Serdang, Malaysia; ykcheah@upm.edu.my; 3Department of Veterinary Clinical Studies, Faculty of Veterinary Medicine, Universiti Putra Malaysia, 43400 Serdang, Malaysia; gayathri@upm.edu.my; 4Institute of Biosciences, Universiti Putra Malaysia, 43400 UPM, Serdang, Malaysia

**Keywords:** breast cancer stem cells, biomarkers, prognosis, microRNAs

## Abstract

Breast cancer is one of the leading causes of cancer-related deaths in women worldwide, and its incidence is on the rise. A small fraction of cancer stem cells was identified within the tumour bulk, which are regarded as cancer-initiating cells, possess self-renewal and propagation potential, and a key driver for tumour heterogeneity and disease progression. Cancer heterogeneity reduces the overall efficacy of chemotherapy and contributes to treatment failure and relapse. The cell-surface and subcellular biomarkers related to breast cancer stem cell (BCSC) phenotypes are increasingly being recognised. These biomarkers are useful for the isolation of BCSCs and can serve as potential therapeutic targets and prognostic tools to monitor treatment responses. Recently, the role of noncoding microRNAs (miRNAs) has extensively been explored as novel biomarker molecules for breast cancer diagnosis and prognosis with high specificity and sensitivity. An in-depth understanding of the biological roles of miRNA in breast carcinogenesis provides insights into the pathways of cancer development and its utility for disease prognostication. This review gives an overview of stem cells, highlights the biomarkers expressed in BCSCs and describes their potential role as prognostic indicators.

## 1. Introduction

Breast cancer is the most commonly diagnosed cancer in women in most parts of the world, with an increasing trend observed. In the United States, the number of new cases of breast cancer is higher than the combined number of lung cancer in both sexes [1]. In Malaysia, it is also the most common cancer among women. The Malaysian National Cancer Registry report in 2019 showed a total of 18,206 new cases between 2012 and 2016, and this accounted for 34.1% of all cancers. The number of cases and the cumulative risk has gradually increased over a decade, from 3.4 (2007) to 3.8 (2016). There was also an increase in the percentage of cases presented at the late stage of the disease between the years 2012 and 2016 (47.9%), as compared to between 2007 and 2011 (43.2%) [2,3].

## 2. Types of Stem Cells

### 2.1. Adult and Embryonic Stem Cells

The pluripotent embryonic stem cells are derived from blastocysts of developing embryos and have the ability to self-renew and differentiate into every type of mature cell and tissue in the body [4,5]. In contrast, adult stem cells are isolated from various organs such as skin, liver, lung and mammary gland. Unlike embryonic stem cells, they have limited differentiation capability and, therefore, are multipotent or unipotent. The progenitor cells will gradually lose their self-renewal ability as they differentiate to produce mature cells [6], with a finite lifespan. In general, stem cells are rare in most of the adult tissues and are difficult to isolate [7]. 

### 2.2. Cancer Stem Cells (CSCs)

CSCs are a subpopulation of cells within malignant tumours. These cells possess stem cell characteristics such as self-renewal, differentiation and the ability to recapitulate the parental tumour when transplanted into a host. They contribute to the growth of tumours, development of chemotherapy resistance and metastasis [8]. In other words, they are the key player of tumour progression and aggressiveness [9]. They express embryonic stem cell factors like sex determining region Y-box 2 (*SOX2*), octamer binding transcription factor 4 (*Oct4*), *Nanog* and DNA (cytosine-5)-methyltransferase 1 (*Dnmt1*) [10,11,12]. The percentage of these cells varies between 0.02% and 25% depending on the tumour type [13]. 

CSCs have been identified in various types of tumours, including breast, brain, colon, leukaemia and lung [10]. Undifferentiated tumours have the highest percentage of CSCs. Conventional cancer treatments are not able to target CSCs, as they are highly resistant, resulting in tumour metastasis and recurrence [9]. 

### 2.3. Breast Cancer Stem Cells (BCSCs)

Al-Hajj et al. (2003) was the first to isolate a subpopulation of cancer cells from human breast tumours based on the expression of the CD44^+^, CD24^−/low^ and epithelial specific antigen (*ESA*)^+^ [14], which was described as BCSCs. Interestingly, these cells were able to form new tumours in immunodeficient mice. Subsequently, Ginestier et al. (2007) reported that high aldehyde dehydrogenase 1 (*ALDH-1*) expression is also a characteristic feature for BCSCs, and it can be used to replace *ESA* as a biomarker [15]. The current accepted markers for BCSCs are CD44^+^, CD24^−/low^ and *ALDH-1*^+^ [16]. The aim of this review is to explore the values of biomarkers of breast cancer stem cells as prognostic factors.

## 3. Prognostic Value of BCSC-Related Biomarkers 

### 3.1. Oct4

*Oct4*, an embryonic stem cell marker, is a transcription factor encoded by the POU Class 5 Homeobox 1 (*POU5F1*) gene. In breast cancer, it was reported to be associated with *ALDH-1* positivity, high Ki-67 proliferative index, high histological tumour grade and an independent prognostic factor. This association was found specifically in the hormone receptor positive breast cancer [17]. Bhatt et al. (2016) reported a possible association between *Oct4* and tamoxifen resistance [18]. *Oct4* overexpression was found to be a poor prognostic factor in various other tumours such as renal cell carcinoma [19], prostate carcinoma [20], bladder carcinoma [21] and cervical carcinoma [22].

### 3.2. Nanog

*Nanog*, an embryonic stem cell marker, is also a biomarker for CSCs. A study showed that its expression was higher in invasive breast cancer, and it was directly associated with tumour size, tumour grade, tumour stage and lymph node metastasis and a poor overall survival [23].

### 3.3. CD133

Croker et al. (2009) found a subpopulation of cells expressing CD133 in human invasive breast cancer cell lines [24], together with the other CSC markers like CD44/CD24 and *ALDH-1*. The number of CD133+ cells was between 1% and 2% in both luminal and human epidermal growth factor receptor 2 (*HER2*)+ cells. Interestingly, in basal-like cell lines, it was up to 80% [25,26]. This suggested that CD133 may be a marker for more aggressive types of breast cancer. 

CD133 messenger RNA (mRNA) and protein expression were also found to be directly correlated with an increasing tumour grade, positive lymph node metastasis, negative progesterone receptor (*PR*) and oestrogen receptor (*ER*), positive *HER2* gene status, advanced tumour, nodes and metastases (TNM) stage and a poor overall survival [27,28]. In addition, CD133 expression in both the cytoplasm and membrane was associated with a shorter survival. A high membrane expression of CD133 was seen in younger ages at the diagnosis of breast cancer [29]. Brugnoli et al. (2013) reported that, by silencing of CD133, it reduced the invasiveness of triple-negative breast cancer (TNBC) [30]. However, some studies described a negative correlation between CD133 expression and poor prognosis. They found an inverse relationship between CD133 levels with the clinical stage of TNBC tumours [31]. Cantile et al. (2013) suggested that a poor prognosis in TNBC is possibly due to a nuclear mislocalisation of CD133, which normally showed membrane positivity [32]. Collina et al. (2015) also reported no statistical association of CD133 expression with TNBC patient survivals [33]. Therefore, the value of CD133 as a prognostic factor is still questionable. 

### 3.4. CD44

In tumours, the epithelial–mesenchymal transition (EMT) is an important step in disease progression. EMT is an embryonic program that is reactivated in tumour cells. It gives additional properties to the tumour cells, such as the ability to invade adjacent tissues and to disseminate under the influence of various cytokines produced by the surrounding stroma cells [34]. The BCSCs can exchange between the epithelial-mesenchymal phenotype and the mesenchymal-epithelial phenotype [35]. ALDH-1 and CD44 are the main markers associated with circulating CSCs, and their detection is able to identify higher metastatic potential [8,36,37].

CD44 is a transmembrane glycoprotein that binds to many extracellular matrix proteins. CD44 mRNA and protein overexpression were observed in the basal subtype of breast cancer. Patients with breast cancer expressing high levels of CD44 have significantly worse overall survival [38]. Although many studies have found that CD44 was associated with a poorer prognosis, the expression status of whether the association is related to a high level or loss of CD44 expression is still questionable. For example, a high CD44 expression was a poor prognostic factor for breast carcinoma [39], lymphoma [40] and thymoma [41], while, in other studies, a loss/reduced CD44 expression was correlated to a poorer prognosis, such as carcinoid tumour [42], gastrointestinal stroma tumour [43], laryngeal carcinoma [44] and lung carcinoma [45]. On the contrary, some studies found that a high level of CD44 was associated with a better prognosis in ovarian cancer and neuroblastoma [46,47].

### 3.5. ALDH-1

ALDH is a family of cytosolic enzymes that oxidises intracellular aldehydes and retinol during the differentiation of rudimentary stem cells [48]. ALDH-1 expression is increased in normal tissue and in many malignancies [49]. Studies on ALDH-1 as an independent prognostic marker in TNBC showed contradicting results. Ma et al. (2017) reported a significant correlation between ALDH-1 expression with tumour size, tumour stage and overall survival [50]. However, Panigoro et al. (2020) found that ALDH-1 can be used as a poor prognostic marker, but it is not an independent prognostic marker [51]. 

### 3.6. Autophagy-Related Genes (Autophagy Protein Light Chain 3 (LC3) and Beclin-1)

Chang et al. (2016) reported that the *LC3*^−^/CD44^+^/CD24^−^ immunophenotype indicated a highly aggressive subgroup of TNBC, which was associated with poor prognosis [52]. *LC3*-negative TNBC has a significant negative association with overall survival [52]. Harmurcu et al. (2018) showed the knockdown of *LC3* and Beclin-1 in MDA-MB-231 and BT-549 TNBC cells, resulting in the inhibition of various proto-oncogenic signalling pathways, such as cyclin D1, urokinase receptor (uPAR)/integrin-β1/ SRC proto-oncogene, nonreceptor tyrosine kinase (*Src*) and poly (ADP-ribose) polymerase 1 (*PARP1*) [53]. It suggested that *LC3* and Beclin-1 are required for cell proliferation, survival, migration and invasion and may contribute to the tumour growth and progression of highly aggressive and metastatic TNBC tumours.

### 3.7. Zinc Finger E-box-Binding Homeobox 1 (ZEB1)

*ZEB1* is an EMT inducer that downregulates E-cadherin and induces the epithelial to mesenchymal transition in breast and other carcinomas [54,55]. *ZEB1* and the loss of E-cadherin were more commonly observed in metaplastic carcinomas than in other subtypes of breast cancer. *ZEB1* expression was identified to be an independent poor prognostic factor for disease-free survival [54]. 

### 3.8. Other Potential Prognostic Markers

El Abbass et al. (2020) evaluated the BCSC marker expressions and the number of mammospheres in cultures of breast cancer tissues and correlated them with patients’ overall survival [56]. The study found that phosphatase and tensin homologue (*PTEN*), phosphoinositide 3-kinase (*PI3K*), protein kinase B (*AKT*), wingless-related integration site (*Wnt*) and β-catenin may play important roles in the development and progression of breast cancer, and they can be used as potential prognostic biomarkers. Lei et al. (2016) performed transcriptome sequencing on BCSCs, breast cancer cells, mammary cells and CD44^+^ mammary cells and found that carbonic anhydrase 12 (*CA12*) was a prognostic biomarker in *HER2*-positive breast cancer [57].

## 4. miRNAs in Stem Cells and Cancer Stem Cells

miRNAs consist of about 18–22 nucleotides and play an important role in virtually all biological processes, including stem cell maintenance, differentiation and development. Using a high-throughput sequencing platform, miRNAs were found in embryonic stem cells, neural stem cells and mesenchymal stem cells [58,59]. The miR-302 cluster comprises of a polycistronic cluster that houses five precursors of miRNAs, i.e., miR-302b, miR302c, miR-302a, miR-302d and miR-367, and is the most well-characterised human embryonic stem cell-specific miRNA [58,60]. Studies have shown that miRNAs are involved in the regulation of breast cancer stem cells [61], pancreatic cancer stem cells [62], colorectal cancer stem cells [63], liver cancer stem cells [64] and cervical cancer stem cells [65]. 

miRNAs that were found to be downregulated in BCSCs are let-7 family, miR-200, miR-30 family, miR-128, miR-34c and miR-16, while those miRNAs that were upregulated consisted of miR-181 and miR-495 [66]. As BCSCs play an important role in the pathogenesis of aggressive phenotypes, miRNAs that affect the maintenance of these cells could be valuable prognostic markers. Table 1 provides a summary of miRNAs that are associated with the regulation of BCSCs.

## 5. Prognostic Value of BCSC-Related MiRNAs

### 5.1. miR-1

Liu et al. (2015) analysed the miRNA expression of 45 breast cancer tissues and serum samples using RT-PCR [81]. They found that the miR-1 expression was significantly lower in basal-like tumours compared to luminal A, luminal B and *HER2*+ tumours. In cases with distant metastasis, the miR-1 level was higher compared to cases without metastasis. They concluded that miR-1 expression was underregulated in BCSCs and inversely related to the aggressiveness of tumours. Zhang et al. (2019) reported a down-regulation of miR-1 in BCSC and it triggered mitophagy of cancer stem cells by binding to the LRPPRC protein and targeting mitochondrial inner membrane organizing system 1 (*MINOS1)* and glycerol-3-phosphate dehydrogenase 2 (*GPD2)* mRNAs [82].

### 5.2. Let-7 miRNA

Let-7 is undetectable in embryonic stem cells, and it became upregulated upon differentiation [58]. In human BCSCs, let-7 suppressed self-renewal and differentiation by targeting a Harvey rat sarcoma viral oncogene homologue (*H-RAS*) and high-mobility group AT-hook 2 (*HMGA2*) [83]. Studies found that the downregulation of let-7 was observed in the CSCs, such as colon cancer and Wilms tumours [84,85].

### 5.3. miR-9 and miR-221

In a study of 206 patients, MiR-9 and miR-221 were found to be associated with stemness features, higher metastatic potential and EMT activation, as well as independently associated with a poorer overall survival and disease-free survival [67].

### 5.4. miR-24

miR-24 was reported to enhance the stemness property of breast cancer cells by expressing antiapoptotic protein BCL21-interacting protein BIM (*BIML*), and its overexpression promoted mammosphere formation in the T-47D, MCF-7 and MDA-MB-231 cell lines. The overexpression of miR-24 also induced the upregulation of *Oct3/4* and *Nanog* stem cell-related genes. Other studies reported miR-24 induced BCSC resistance against cisplatin [69,92].

### 5.5. miR-27a

Tang et al. (2014) demonstrated that BCSCs treated with vascular endothelial growth factor (*VEGF*) increased miR-27a and suggested that the interaction between *VEGF* and miR-27a promoted angiogenesis and metastasis [70]. miR-27a was upregulated in breast cancer and was correlated with tumour size, lymph node metastasis and distant metastasis. They suggested that a high level of miR-27a is associated with overall poor survival [71].

### 5.6. miR-125a

A study showed that increasing the expression of miR-125a led to an increase in BCSC populations in MCF12A cells. In sphere-forming assays, the addition of miR-125a enhanced the sphere-forming ability by about 1.5-fold compared to mock-treated control cells. In contrast, miR-125a inhibition resulted in a decreased stem cells population and a decrease in sphere formation. They also found miR-125a overexpressing MCF12A spheres promoted *SOX2* expression, a stem cell marker [72].

### 5.7. miR-128

Breast cancer stem cells and breast cancer cells downregulated miR-128-3p. A study showed miR-128-3p overexpression inhibited the proliferation, migration, invasion, self-renewal and tumorigenicity of BCSCs via the downregulation of serine/threonine-protein kinase Nek2 (*NEK2*), a gene involved in cell division. The study suggested that miR-128-3p inhibits the stem-like cell features of BCSCs [86].

### 5.8. miR-142

Isobe et al. (2014) reported that the ectopic expression of miR-142 in normal mouse mammary stem cells led to the formation of hyperproliferative mammary glands [73]. By knockdown miR-142, it suppressed the organoid formation by BCSCs and reduced the tumour growth initiated by BCSCs. miR-142 regulated the properties of BCSC by, in part, activating the *Wnt*-signalling pathway and miR-150. The study suggested that miR-142 is frequently upregulated in BCSCs.

### 5.9. miR-200

Shimono et al. (2009) isolated BCSCs directly from breast cancer samples and identified a subset of miRNAs that were differentially expressed between BCSCs and nontumourigenic cancer cells [75]. These miRNAs are the miR-200 family, let-7, miR-1 and miR-27. The miR-200 family comprises five members that are located in two clusters: the first cluster is on human chromosome 1 (miR−200a, miR−200b and miR−429), and the second cluster is on human chromosome 12 (miR−200c and miR−141) [76]. The miR-200 family modulates the self-renewal ability of CSCs by targeting the B-lymphoma Mo-MLV insertion region 1 homologue (*BMI-1*) and SUZ12 polycomb repressive complex 2 subunit (*SUZ12)* [77]. *BMI-1* regulates the self-renewal and differentiation of several types of stem cells [78].

### 5.10. miR-205

Zhang et al. (2020) reported that miR-205 expression was reduced in CD44+/CD24^−^/low BCSCs compared with non-BCSCs [87]. The overexpression of miR-205 in the MB-231 cell line led to a reduced CD44^+^/CD24^−/low^ population. They suggested that miR-205 could inhibit breast cancer malignancy by regulating *RUNX2*, and miR-205 is a tumour suppressor during breast cancer development.

### 5.11. miR-210

Tang et al. (2018) reported that the upregulation of miR-210 induced by a hypoxic microenvironment promoted breast cancer stem cell metastasis, proliferation and self-renewal by targeting E-cadherin [79]. They also found that miR-210 was upregulated in culture MCF-7 spheroid cells, which were high in BCSCs compared with MCF-7 parental cells. A high miR-210 expression was also detected in CD44^+^/CD24^−^ MCF-7 cells and human CD44^+^/CD24^−^ breast cancer cells.

### 5.12. miR-495

MiR-495 expression was found to be upregulated in BCSCs. The overexpression of miR-495 in breast cancer cells increased the colony-forming capacity in vitro and tumorigenesis in mice. The study also reported that miR-495 suppressed E-cadherin expression to promote cell invasion, while it inhibited the regulation in development and DNA damage responses 1 (*REDD1*) expression in a hypoxia environment to enhance cell proliferation. They concluded that miR-495 contributed to BCSC properties [80].

### 5.13. miR-590-5p

Zhou et al. (2017) evaluated the expression of miR-590-5p by real-time polymerase chain reaction (RT-PCR) on 49 breast cancer patients and compared them with nontumorous tissue and found that miR-590-5p expression was reduced in breast cancer tissues [88]. The overexpression of miR-590-5p significantly decreased the ALDH1-positive cell population. They also reported that miR-590-5p significantly downregulated SOX2 protein expression and vice versa. They concluded that miR-590-5p inhibited breast cancer cell stemness through targeting SOX2.

### 5.14. miR-628

A transfection study using miRNA mimics showed miR-628 suppressed the migration and invasiveness of BCSCs of MDA-MB-231 and MCF-7 cells by targeting the Son of sevenless homologue 1 (*SOS1*), which attenuated snails and vimentin, while enhancing E-cadherin activity. They also found that miR-628 was downregulated in bone metastatic breast cancer cells [89].

### 5.15. miR-638

Lin et al. (2020) investigated the relationship between miR-638 (a tumour suppressor gene) and E2F transcription factor 2 (*E2F2*) in BCSCs [90]. In breast cancer, miR-638 was downregulated, while *E2F2* was elevated. The overexpression of miR-638 resulted in a reduction of CD24^−^/CD44^+^ cells and the stem cell markers like *SOX2* and *Oct4*. In addition, miR-638 overexpression inhibited the abilities of BSCSs to self-renew, proliferate and invade. Overall, miR-638 overexpression caused a reduction in tumour growth.

### 5.16. miR-760

Han et al. (2016) evaluated the expression of miR-760 in the BT-549 cell line and MCF-7 cell line [91]. They found that the BT-549 cell line had a lower expression of miR-760 and a high level of *Nanog* expression. The BT-549 cell line had a greater number of CD44^+^/CD24^−/low^ subpopulations. The overexpression of miR-760 in the BT-549 cell line resulted in reduced CD44^+^/CD24^−/low^ cells and inhibited cell proliferation and migration by downregulating *Nanog*.

### 5.17. Other miRNAs

Several studies reported the association of miRNAs with disease progression. miR-33b targeted *HMGA2*, *SALL4* and *Twist1* and inhibited the stemness, migration and invasion of metastatic breast cancer cells [74]. miR-199a promoted BCSC propagation, tumour initiation and metastasis by the suppression of FOXP2 [93]. miR-20a enhanced the metastatic abilities of BCSCs by conferring the resistance of BCSCs to natural killer (NK) cell cytotoxicity [68].

## 6. Conclusions

In summary, these miRNAs exert their effects by regulating three main pathways: (1) by maintaining the stemness properties of BCSCs via upregulating *Oct4*; *SOX2* and *Nanog* expressions (miR-24, miR-125a, miR-590-5p and miR-760); (2) by regulating the proliferation and self-renewal of BCSCs via the cell cycle and cell division genes like *E2F2*; *HRAS*; *NEK2* and *RUNX2* (let-7, miR-128, miR-205 and miR-638) and (3) by suppressing E-cadherin, a cell-cell adhesion molecule, to promote proliferation; invasion and metastasis (miR-210, miR-495 and miR-628) (Figure 1). As miRNAs can be isolated in the serum as membrane-bound vesicles, the identification of circulating BCSC-related miRNAs could be used as a form of prognostication of breast cancer.

In the era of precision medicine, gene-targeting, immunomodulatory and cell-based therapies will be potential promising tools for breast cancer treatment. miRNA mimics and inhibitors could be used as therapeutic agents. Various studies have described the roles of miRNAs in the maintenance of BCSCs and the regulation of the stemness properties of these cells. The post-transcriptional regulation of miRNA is complex, as a single miRNA could target multiple mRNAs, while multiple miRNAs could target a single mRNA. Tumour growth and progression involve complex interactions between cancer cells, CSCs, inflammatory cells and fibroblasts within the tumour microenvironment. The development of miRNA-based therapy required intensive research to unravel the complex regulation of miRNAs in CSCs. Exploring the regulation of these miRNAs will further advance our knowledge of the roles of human BCSCs in tumour progression and their prognostic value in breast cancer.

## Figures and Tables

**Figure 1 diagnostics-10-00721-f001:**
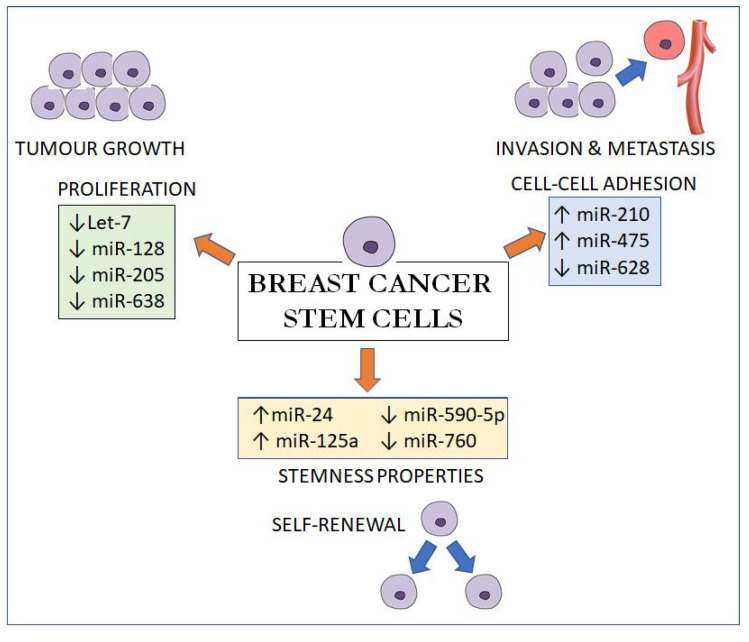
MicroRNA effects on breast cancer stem cells on proliferation, cell-cell adhesion and stemness properties.

**Table 1 diagnostics-10-00721-t001:** MicroRNAs associated with the regulation of breast cancer stem cells.

	Upregulated miRNAs	Remarks	References
1	miR-9/221	Increases tumour size, poor differentiation, lymph node metastasis and lower overall survival	[67]
2	miR-20a	Downregulates MICA/MICB to promote the resistance of BCSCs to NK cell cytotoxicity	[68]
3	miR-24	Induces the upregulation of *Oct3-4* and *Nanog*, as well as the expression of antiapoptotic BIML	[69]
4	miR-27a	VEGF increases miR-27a and, together, promote angiogenesis and tumour metastasis	[70,71]
5	miR-125a	Promotes *SOX2* expression	[72]
6	miR-142	Regulates BCSCs properties by in part activating the WNT signalling pathway	[73]
7	miR-199a	Suppresses *FOXP2*	[74]
8	miR-200	Modulates CSC self-renewal via *BMI-1* and *SUZ12*	[75,76,77,78]
9	miR-210	miR-210 is reduced in the hypoxic environment and, in turn, promotes BCSC proliferation, self-renewal and metastasis by targeting E-cadherin	[79]
10	miR-495	Promotes cell invasion by suppressing E-cadherin	[80]
	**Downregulated miRNAs**		
1	miR-1	Increases in basal-like subtype Triggered mitophagy of cancer stem cells by targeting *MINOS1* and *GPD2* mRNAs	[81,82]
2	let-7	Suppresses self-renewal and differentiation by targeting *HRAS* and *HMGA2*	[83,84,85]
3	miR-33b	Targets *HMGA2*, *SALL4* and *TWIST1*	[74]
4	miR-128	Inhibits proliferation, migration, invasion, self-renewal and tumorigenesis by downregulating *NEK2*	[86]
5	miR-205	A tumour suppressor that inhibits breast cancer malignancy by regulating *RUNX2*	[87]
6	miR-590-5p	Inhibits breast cancer cell stemness by downregulating *SOX2*	[88]
7	miR-628	Suppresses BCSCs invasiveness by targeting *SOS1* and enhancing E-cadherin	[89]
8	miR-638	Reciprocal effect with E2F2 in BCSCs, inhibiting self-renewal, proliferation and invasion	[90]
9	miR-760	Inhibits cell proliferation and migration by downregulating Nanog	[91]

B lymphoma Mo-MLV insertion region 1 homolog - BMI1; Breast Cancer Stem Cells – BCSC; E2F transcription factor 2 - E2F2; Forkhead box protein P2 - FOXP2; Glycerol-3-phosphate dehydrogenase 2 - GPD2; High-mobility group AT-hook 2- HMGA2; MHC Class I Polypeptide-Related Sequence A/B - MICA/MICB; Mitochondrial inner membrane organizing system 1 - MINOS1; Runt-related transcription factor 2 - RUNX2; Sal-like protein 4- SALL4; Serine/threonine-protein kinase Nek2 - NEK2; Son of sevenless homolog 1 - SOS1; SRY (sex determining region Y)-box 2 - SOX2; SUZ12 polycomb repressive complex 2 subunit - SUZ12; Twist-related protein 1- TWIST1; Vascular endothelial growth factor – VEGF.

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
