# Peer review of "The Role of Breast Cancer Stem Cell-Related Biomarkers as Prognostic Factors"

_diagnostics, 2020, doi:10.3390/diagnostics10090721_

Round 1

Reviewer 1 Report

Manuscript ID: diagnostics-917231 Type of manuscript: Review

Title: The Role of Breast Cancer Stem Cells-Related MicroRNAs as Prognostic Biomarkers

Authors: Clarence Ching Huat Ko, Wai Kit Chia, Gayathri Thevi Selvarajah, Yoke Kqueen Cheah, Yin Ping Wong, Geok Chin Tan *

Submitted to section: Pathology and Molecular Diagnostics

First, I must say that the topic is remarkably interesting and relevant in the field of breast cancer. It is complex and debatable and further studies in this field are urgently needed. Du to the difference in opinion by different researchers, it is important to write a review like the one the authors of this manuscript are trying to achieve.

There is a major problem with the references. They have obviously been mixed. It seems that they have been skewed already from quite early in the manuscript. It starts already from reference number 14. By this I mean for example reference number 38 should be reference number 39, reference number 39 should be reference number 40, and so forth. This is indeed annoying when trying to follow the content of the text. It takes focus away from the meaning of the text. This should be sorted out before any publication is considered.

The different chapters and topics are short and to the point. There is a relatively good description of the different biomarkers and the microRNAs. It is questionable why there is only one reference to each biomarker and microRNA. Clearly there has been done more research on the individual markers. Obviously, one cannot refer to all the work that has been done, but a few more relevant references would increase the knowledge of its function.

Table 1 is good and descriptive. It would be better if the references to the different microRNAs are mentioned in the table as well as in the text.

Figure 1 is the best part of the whole review. It is illustrative and easily read and understood. It increases the value of the review significantly

Author Response

RESPONSE TO REVIEWER 1

Reviewer 1

First, I must say that the topic is remarkably interesting and relevant in the field of breast cancer. It is complex and debatable and further studies in this field are urgently needed. Due to the difference in opinion by different researchers, it is important to write a review like the one the authors of this manuscript are trying to achieve.

Point 1: There is a major problem with the references. They have obviously been mixed. It seems that they have been skewed already from quite early in the manuscript. It starts already from reference number 14. By this I mean for example reference number 38 should be reference number 39, reference number 39 should be reference number 40, and so forth. This is indeed annoying when trying to follow the content of the text. It takes focus away from the meaning of the text. This should be sorted out before any publication is considered.

Response 1:

Thank you for pointing out the mistake. We have accidentally included the link in reference 3 as reference 4. It is now rectified. Page 10, line 318.

Point 2: The different chapters and topics are short and to the point. There is a relatively good description of the different biomarkers and the microRNAs. It is questionable why there is only one reference to each biomarker and microRNA. Clearly there has been done more research on the individual markers. Obviously, one cannot refer to all the work that has been done, but a few more relevant references would increase the knowledge of its function.

Response 2:

To the best of our knowledge, we have tried our best to include all relevant published results in this review.

Point 3: Table 1 is good and descriptive. It would be better if the references to the different microRNAs are mentioned in the table as well as in the text.

Response 3:

This is a very good suggestion. We have added the references into the table. A new column is added to the table. Page 5.

Point 4: Figure 1 is the best part of the whole review. It is illustrative and easily read and understood. It increases the value of the review significantly

Response 4:

Thank you for the comment.

Reviewer 2 Report

Ching Huat Ko et al present a review on Cancer Stem Cells-Related MicroRNAs as Prognostic Biomarkers. The abstract mentions breast cancer, cancer stem cells, and miRNAs; yet, at the end, it suggests a broader scope, citing only biomarkers expressed in BCSCs (line 28). This idea is stated again in lines 71-72. Although seemingly trivial, this point summarizes my main concern about this paper. 

The idea of reviewing miRNAs related to cancer stem cells is very interesting, and the information on this topic that Ching Huat Ko and colleagues present is recent and reasonably thorough. That said, I find the structure of this review very confusing:

First. The review is clearly aimed as BCSC-related miRNAs, so I do not understand the purpose of Section 3 (Lines 74 to 154). Sure, miRNAs are not the only BCSC markers,  but the title suggests a narrow scope review. The authors make this a bit obscure by mentioning ‘accepted’ markers (Line 70) and mentioning CD44+, CD24−/low, and ALDH-1+ categorically without explaining the criteria for acceptance or providing a citation for this statement. Are the other markers not accepted? Are miRNAs not accepted as markers?

I suggest eliminating Section 3 and focus on BCSC-related miRNAs, elaborating more on them as a group of regulators, besides only mentioning that they are involved in the regulation of breast cancer stem cells (Lines 161-162).

Second. Section 5 (Lines 169 to 275) is subdivided by individual miRNAs, in a numerical order. I don not find this order useful or informative, as miRNA numbers are assigned arbitrarily; besides, this order does not correspond to that shown in Table 1 or Figure 1. Moreover,  I am not sure that data from a a single paper merits a subsection in a review (e.g. sections 5.5, 5.6, or 5.7) . 

I suggest organizing the information about the various miRNAs in either two (up- and down-regulated) or three (proliferation- , invasion- and metastasis-inducing) sections, so that there is cohesion between the text and either the table or figure. From a personal perspective, I would take the three-section path and focus on the function.

Finally, the discussion section summarizes the presented information related to oncogenic processes, a distinction that is not mentioned before (see above), and then suggests that ‘miRNA mimics and inhibitors could be used as therapeutic agents’ (Lines 288-289). While true, this idea does not refer back to the original aim of the review, BCSC-related MicroRNAs as prognostic biomarkers. Coming back to the usefulness of miRNAs as biomarkers in the discussion would render this review more cohesive.

Minor points

Please provide citations to back up the statement  “CSCs have been identified in various types of tumours including breast, brain, colon, leukaemia and lung.” (Lines 60-61)

Please clarify the relevance of the miR-302 cluster (lines 159-161), as it seems to be exclusive to embryonic and not cancer stem cells.

Line 175 - perhaps the authors meant the ‘aggressiveness’ of the tumor 

Citations for references are moved one position  from 55 (Line 148) on, so there are 91 in-text citations and a reference list of 92 items.

Author Response

RESPONSE TO REVIEWER 2

Reviewer 2

Ching Huat Ko et al present a review on Cancer Stem Cells-Related MicroRNAs as Prognostic Biomarkers. The abstract mentions breast cancer, cancer stem cells, and miRNAs; yet, at the end, it suggests a broader scope, citing only biomarkers expressed in BCSCs (line 28). This idea is stated again in lines 71-72. Although seemingly trivial, this point summarizes my main concern about this paper. 

The idea of reviewing miRNAs related to cancer stem cells is very interesting, and the information on this topic that Ching Huat Ko and colleagues present is recent and reasonably thorough. That said, I find the structure of this review very confusing:

Point 1: First. The review is clearly aimed as BCSC-related miRNAs, so I do not understand the purpose of Section 3 (Lines 74 to 154). Sure, miRNAs are not the only BCSC markers, but the title suggests a narrow scope review. The authors make this a bit obscure by mentioning ‘accepted’ markers (Line 70) and mentioning CD44+, CD24−/low, and ALDH-1+ categorically without explaining the criteria for acceptance or providing a citation for this statement. Are the other markers not accepted? Are miRNAs not accepted as markers?

I suggest eliminating Section 3 and focus on BCSC-related miRNAs, elaborating more on them as a group of regulators, besides only mentioning that they are involved in the regulation of breast cancer stem cells (Lines 161-162).

Response 1:

We agree that the title is focus mainly on microRNA, while the abstract our discussion was broader. In response to this, the title has been changed to “The Role of Breast Cancer Stem-Cells-Related Biomarkers as Prognostic Factors” to include all other markers.

Page 1, line 2-3.

CD44+, CD24−/low, and ALDH-1+ are the most established and specific markers of breast cancer stem cells. The other cell surface markers that were mentioned in the review are mostly embryonic stem cell markers. This review intends to explore the role of published markers related breast cancer stem cells as prognostic factor. A new reference was added [16] as suggested. See page 2, line 71 and page 10, line 343. As a result of the addition of a new reference, the in-text citation from reference no. 17 onwards & the reference list in page 10-14 were also updated accordingly.

[16] Ricardo S, Vieira AF, Gerhard R, et al. Breast cancer stem cell markers CD44, CD24 and ALDH1: expression distribution within intrinsic molecular subtype. J Clin Pathol. 2011;64(11):937-946.

Are miRNAs not accepted as markers?

This is an interesting question. Currently, there are no acceptable miRNA marker for breast cancer stem cells.

It will be exciting to identify a specific miRNA marker for breast cancer stem cells. There are only a handful of recognised miRNA markers of cancer. One example is embryonal tumour with multi-layered rosettes (ETMR) with C13MC (Cluster 13 microRNA). (See reference Pei et al. 2019)

Pei YC, Huang GH, Yao XH, Bian XW, Li F, Xiang Y, Yang L, Lv SQ, Liu J. Embryonal tumor with multilayered rosettes, C19MC-altered (ETMR): a newly defined pediatric brain tumor. Int J Clin Exp Pathol. 2019 Aug 1;12(8):3156-3163.

Point 2: Second. Section 5 (Lines 169 to 275) is subdivided by individual miRNAs, in a numerical order. I do not find this order useful or informative, as miRNA numbers are assigned arbitrarily; besides, this order does not correspond to that shown in Table 1 or Figure 1. Moreover, I am not sure that data from a a single paper merits a subsection in a review (e.g. sections 5.5, 5.6, or 5.7). 

I suggest organizing the information about the various miRNAs in either two (up- and down-regulated) or three (proliferation-, invasion- and metastasis-inducing) sections, so that there is cohesion between the text and either the table or figure. From a personal perspective, I would take the three-section path and focus on the function.

Response 2:

Thank you for the comments. As the table has summarise the miRNAs according to the up and down-regulation and roles in breast cancer stem cells. I personally think it would be an advantage to have both presentations, i.e. 1) Based on individual miRNAs in numerical order and 2) Based on the up and down-regulation. This will be clear to the readers and they could choose to search for the miRNAs in either ways. I believe this is an advantage. In addition, the figure has also summarised the effect of miRNAs on breast cancer stem cells. However, if the editor/reviewer think otherwise, I would gladly change it.

Point 3: Finally, the discussion section summarizes the presented information related to oncogenic processes, a distinction that is not mentioned before (see above), and then suggests that ‘miRNA mimics and inhibitors could be used as therapeutic agents’ (Lines 288-289). While true, this idea does not refer back to the original aim of the review, BCSC-related MicroRNAs as prognostic biomarkers. Coming back to the usefulness of miRNAs as biomarkers in the discussion would render this review more cohesive.

Response 3:

Thank you for the comments.

We discussed about the potential usage of miRNA mimics and inhibitors as therapeutic agents. Although it is unrelated to the prognostic point of view, it is an important food for thought as tumour with very poor prognosis will require new form of therapy, in particularly in the form of precision medicine, which we believe should be mentioned.

In response to your comment, a new statement was added in the conclusion.

Page 8, line 282-284.

Point 4: Minor points

Please provide citations to back up the statement “CSCs have been identified in various types of tumours including breast, brain, colon, leukaemia and lung.” (Lines 60-61)

Response 4:

Reference added [10]

This is a review article on cancer stem cells and their potential targeted therapies.

Reference 10: Desai A, Yan Y, Gerson SL. Concise Reviews: Cancer Stem Cell Targeted Therapies: Toward Clinical Success. Stem Cells Transl Med. 2019 Jan;8(1):75-81.

Point 5: Please clarify the relevance of the miR-302 cluster (lines 159-161), as it seems to be exclusive to embryonic and not cancer stem cells.

Response 5:

It is correctly pointed out that miR-302 cluster is mainly an embryonic stem cells specific miRNA. The purpose of this paragraph is to provide a general idea that miRNA can be specific in stem cells. Therefore, it may be possible to identify a miRNA specific to breast cancer stem cells.

Point 6: Line 175 - perhaps the authors meant the ‘aggressiveness’ of the tumor 

Response 6:

Thank you for pointing out the mistake. It has been corrected.

Point 7: Citations for references are moved one position from 55 (Line 148) on, so there are 91 in-text citations and a reference list of 92 items.

Response 7:

Thank you for pointing out the mistake. We have accidentally included the link in reference 3 as reference 4. It is now rectified. Page 10, line 318.

Round 2

Reviewer 1 Report

Interesting paper. Ready to be published